# The activity of TRAF RING homo- and heterodimers is regulated by zinc finger 1

Adam J. Middleton [1], Rhesa Budhidarmo[1], Anubrita Das[1], Jingyi Zhu[1], Martina Foglizzo [1]
Peter D. Mace [1] & Catherine L. Day [1]

Ubiquitin chains linked through lysine63 (K63) play a critical role in inflammatory signalling. Following ligand engagement of immune receptors, the RING E3 ligase TRAF6 builds K63-linked chains together with the heterodimeric E2 enzyme Ubc13-Uev1A. Dimerisation of the TRAF6 RING domain is essential for the assembly of K63-linked ubiquitin chains. Here, we show that TRAF6 RING dimers form a catalytic complex where one RING interacts with a Ubc13~Ubiquitin conjugate, while the zinc finger 1 (ZF1) domain and linker-helix of the opposing monomer contact ubiquitin. The RING dimer interface is conserved across TRAFs and we also show that TRAF5–TRAF6 heterodimers form. Importantly, TRAF5 can provide ZF1, enabling ubiquitin transfer from a TRAF6-bound Ubc13 conjugate. Our study explains the dependence of activity on TRAF RING dimers, and suggests that both homo- and hetero-dimers mediated by TRAF RING domains have the capacity to synthesise ubiquitin chains.

---

[1] Department of Biochemistry, School of Biomedical Sciences, University of Otago, Dunedin 9054, New Zealand. Rhesa Budhidarmo and Anubrita Das contributed equally to this work. Correspondence and requests for materials should be addressed to C.L.D. (email: catherine.day@otago.ac.nz)

Many immune signalling pathways rely on the synthesis of ubiquitin chains. Non-degradative ubiquitin chains have important roles in determining the strength, duration and type of inflammatory response by functioning as molecular glue to stabilise signalling complexes. The extent of ubiquitin chain synthesis following cytokine engagement determines whether downstream effector molecules are recruited and activated. One E3 ligase with a critical role in many immune signalling pathways is TNF receptor-associated factor 6 (TRAF6). TRAF6 was initially identified because of its requirement for interleukin1-receptor (IL1-R)-mediated activation of NF-κB[1], but is now known to play key roles in multiple signalling pathways that control immunoregulatory functions[2, 3]. This is because ubiquitin chains that are synthesised by TRAF6 serve as a platform for the activation of downstream kinases such as TAK1 and Akt[4, 5].

Befitting its central role in signalling, disruption of TRAF6 function has been linked to cancer and inflammatory disorders. Amplification of TRAF6 is associated with lung carcinoma[6], and poor prognosis for head and neck cancers[7]. Overexpression of TRAF6 predicts a poor response to chemotherapy and radiotherapy for colorectal cancer patients, with molecular studies suggesting this is because TRAF6 regulates mitochondrial

translocation of p53[8]. It has also been suggested that chronic activation of TRAF6 is associated with aberrant splicing in haemopoietic cells in myelodysplastic syndromes[9].

TRAF6 belongs to the TRAF family of proteins (TRAF1–7), which are defined by the presence of a TRAF-C/MATH domain and a TRAF-N or coiled coil (CC) domain[10]. These domains are responsible for trimerisation and receptor binding[10, 11]. At their N-termini TRAF2–7 have a RING domain, followed by a series of zinc finger (ZF) domains[10] (Fig. 1a). The RING domain is common to many ubiquitin E3 ligases and, when dimeric, confers on TRAF6 its ubiquitin chain-building activity[12]. In general, RING E3 ligases bring together a substrate and a ubiquitin-conjugated E2 enzyme resulting in transfer of ubiquitin to a substrate lysine residue[13, 14]. There are ~30 E2s that assemble ubiquitin chains of different types. TRAF6 preferentially builds Lys63-linked ubiquitin chains by virtue of its interaction with the Ubc13-Uev1A (also called Ube2N-Ube2V1) E2 complex following engagement of receptors[15]. For other TRAFs, the E2 partners are still to be defined, and in fact TRAF2 contains an insertion in its RING domain that appears to abrogate E2 binding[16].

The RING domain not only recruits the E2~ubiquitin (E2~Ub) conjugate, but also enhances the rate of ubiquitin transfer from the E2 to the substrate[17, 18]. Recent studies have shown RING E3s

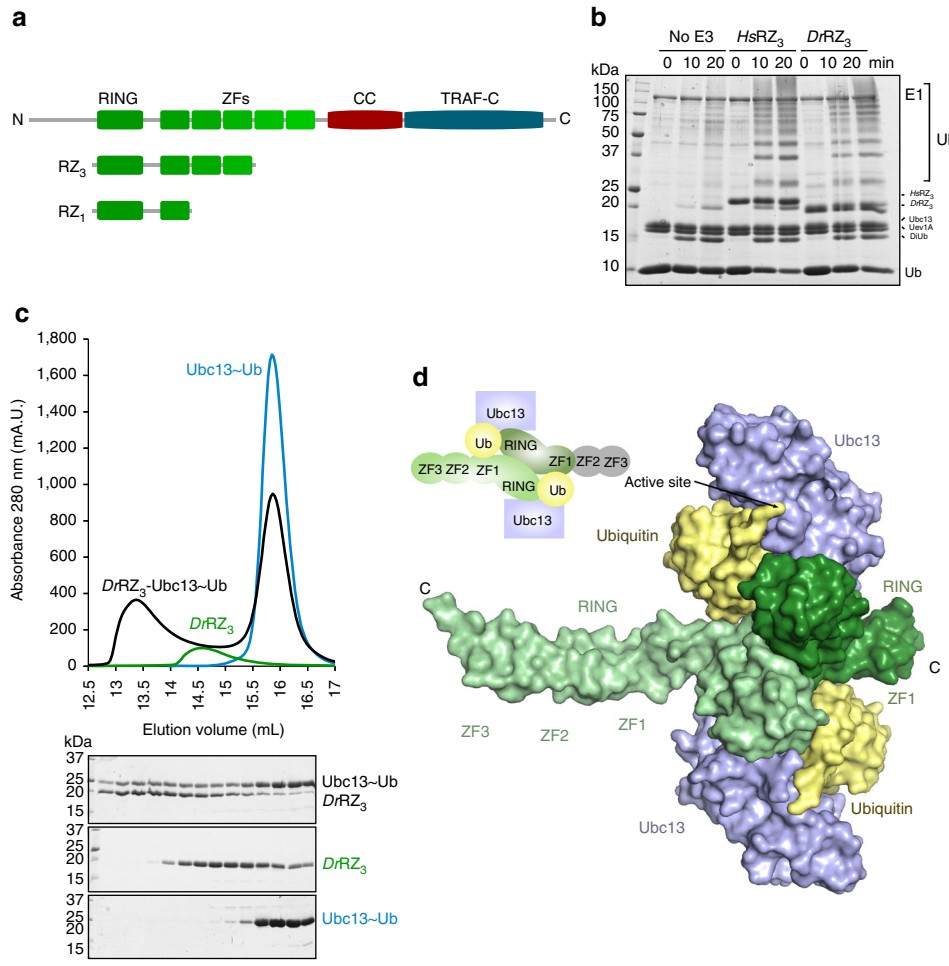

**Fig. 1** Structural characterisation of TRAF6 in complex with a Ubc13~Ub conjugate. **a** Schematic showing the domains of TRAF6, with the RZ₃ and RZ₁ constructs indicated below. ZF: zinc finger, CC: coiled-coil. **b** Multi-turnover activity assay comparing *H. sapiens* and *D. rerio* RZ₃ TRAF6. Each form of TRAF6 RZ₃ was incubated with E1, Ubc13-Uev1A and ubiquitin at 37 °C for the indicated times. **c** Analytical size exclusion profile of 250 μM *Dr*RZ₃ and the isopeptide-linked Ubc13–ubiquitin conjugate (Ubc13-Ub) alone, or a mixture of the two were separated using a Superdex 200 10/300 column. Equivalent fractions from each run were resolved by 16% SDS-PAGE. **d** Crystal structure of the complex of *Dr*RZ₃ and Ubc13-Ub showing the asymmetric unit. The two TRAF6 monomers are in shades of green, ubiquitin is yellow and Ubc13 is blue. The disordered ZF2 and ZF3 are indicated in grey in the schematic

achieve this by binding the E2 conjugated ubiquitin so that the flexible linker between ubiquitin and the E2 is bound to a groove on the E2[19–24]. While some RING-ubiquitin contacts are conserved[14], the additional interactions that prime the conjugate for catalysis differ[19, 20, 22–25]. Like TRAF6, the activity of many RING domains depends on their dimerisation because one RING domain binds the E2, while residues from the C terminus of the other RING protomer make essential contacts with ubiquitin[19, 21–26]. However, TRAF RING domains do not contain a structural equivalent to the C-terminal tail that is essential for many dimeric RING domains[19, 21]. Therefore, the mechanism by which TRAFs stabilise the E2~Ub conjugate in the active (closed) conformation remains uncertain. Furthermore, many receptor complexes contain at least two different TRAF proteins. This raises the possibility that TRAF heterodimers could be functional —as observed for other RING E3 ligases such as MDM2/MDMX and BRCA1/BARD1[27, 28].

In TRAF6, the RING together with ZF1 is required for E3 ligase activity[12, 29]. To identify the essential features that regulate ubiquitin chain assembly by TRAF6 we have determined structures of the RING and ZF domains of TRAF6 bound to the Ubc13~Ub conjugate in two crystal forms at 3.4 and 3.9 Å. Both structures show that the Ubc13~Ub conjugate contacts both protomers of the TRAF6 dimer. Importantly, ZF1 and the helix that connects it to the RING domain contact ubiquitin and enhance the rate of ubiquitin transfer. The structure and biochemical assays explain why RING dimerisation is required for assembly of ubiquitin chains by TRAF6, and account for the importance of the ZF domains for activity. Furthermore, we also show that the conserved RING dimer interface facilitates formation of TRAF heterodimers that are functional E3 ubiquitin ligases.

## Results

**Structure of the Ubc13~Ub conjugate bound to the TRAF6 dimer.** To understand the molecular basis of Lys63 chain synthesis by TRAF6 we sought to obtain the structure of TRAF6 bound to the Ubc13~Ub conjugate. The structure of the N-terminal RING domain and the first three ZFs of human TRAF6 (residues 50–211) has been reported previously[12] and we initially used a comparable human construct, referred to here as RZ$_3$ (HsRZ$_3$). Using GST-pulldown assays and analytical size-exclusion chromatography, we established that HsRZ$_3$ preferentially binds to the Ubc13~Ub conjugate (Supplementary Fig. 1). Because TRAF6 HsRZ$_3$ formed a stable complex with the isopeptide-linked Ubc13~Ub conjugate, we purified the complex and set up crystal trials. We obtained crystals of the HsRZ$_3$–Ubc13~Ub complex, but despite extensive optimisation these diffracted to very low resolution. In an attempt to improve crystal quality, we expressed and purified the equivalent RZ$_3$ construct from Danio rerio TRAF6 (DrRZ$_3$). The sequence of the RING domain and first ZF are highly conserved between DrRZ$_3$ and HsRZ$_3$ (Supplementary Fig. 2a), and together with Ubc13 and Uev1A, DrRZ$_3$ and HsRZ$_3$ promoted the comparable synthesis of ubiquitin chains (Fig. 1b). Additionally, DrRZ$_3$ formed a stable complex with the Ubc13~Ub conjugate (Fig. 1c) that could be crystallised and diffracted to 3.9 Å (Table 1). The structure of the complex was readily solved by molecular replacement (Supplementary Fig. 2b).

In the structure, each asymmetric unit contains a single TRAF6 RZ$_3$ dimer bound to two Ubc13~Ub conjugate molecules (Fig. 1d). The central RING dimer is very similar to the previously reported HsTRAF6 RZ$_3$ dimer structure[12], and an overlay of the RING domains (residues 62–122) has a main-chain RMSD of 1.0 Å. The contacts between the conjugate and TRAF6

**Table 1 Data collection and refinement statistics**

|  | DrRZ$_3$ (5VOO) | DrRZ$_1$ (5VNZ) |
|---|---|---|
| *Data collection* |  |  |
| Space group | P4$_2$2$_1$2 | C222$_1$ |
| Cell dimensions |  |  |
| $a, b, c$ (Å) | 181.11, 181.11, 97.41 | 138.36, 170.55, 97.31 |
| $\alpha, \beta, \gamma$ (°) | 90, 90, 90 | 90, 90, 90 |
| Resolution (Å) | 49.4–3.90 (4.36–3.90)$^a$ | 42.3–3.40 (3.68–3.40) |
| $R_{merge}$ | 0.167 (0.637) | 0.119 (1.43) |
| $I/\sigma(I)$ | 7.9 (2.6) | 9.7 (1.2) |
| $CC_{1/2}$ | 0.993 (0.802) | 0.997 (0.640) |
| Completeness (%) | 97.9 (98.6) | 99.6 (98.3) |
| Redundancy | 4.9 (4.9) | 7.1 (7.1) |
| *Refinement* |  |  |
| Resolution (Å) | 3.90 | 3.40 |
| No. reflections | 14,872 | 16,020 |
| $R_{work}/R_{free}$ | 0.252 / 0.299 | 0.237 / 0.294 |
| No. of atoms |  |  |
| Protein | 5660 | 5321 |
| Ion (Zn$^{2+}$) | 8 | 6 |
| Water | 0 | 0 |
| B factors |  |  |
| Protein | 113.0 | 159.5 |
| Ligand/ion | 94.21 | 131.6 |
| R.m.s. deviations |  |  |
| Bond lengths (Å) | 0.023 | 0.007 |
| Bond angles (°) | 1.54 | 1.06 |

Each structure was determined from a single crystal
$^a$Values for the highest-resolution shell are shown in parentheses

are comparable on both sides of the dimer, but the structure is asymmetric because there is no density for ZF2 and ZF3 in one molecule of TRAF6. There was no evidence of any proteolysis, and a large solvent channel occupies the space where ZF2 and ZF3 are expected, suggesting that the two missing ZFs are highly mobile in the absence of any crystal contacts. We therefore expressed and purified a truncated construct that contained the RING and just ZF1 of D. rerio TRAF6 (denoted DrRZ$_1$). RZ$_1$ and RZ$_3$ had a comparable level of activity (Supplementary Fig. 2c) suggesting that the shorter construct contains the key determinants of activity.

TRAF6 RZ$_1$ formed a stable complex with Ubc13~Ub that allowed purification and crystallisation of the complex. The structure was solved at 3.4 Å and contains two 1:1 DrRZ$_1$-Ubc13~Ub complexes in the asymmetric unit (Supplementary Fig. 2d). Each DrRZ$_1$-Ubc13~Ub conjugate complex sits on a twofold axis and as a result the crystal contains two distinct symmetrical complexes, which have a centrally positioned TRAF6 RING dimer associated with two Ubc13~Ub conjugates (Supplementary Fig. 2e). The α-carbons of the two symmetrical RZ$_1$ complexes overlay with an RMSD of 0.4 Å, and both are similar to the RZ$_3$ complex, with an RMSD of 0.9 and 1.5 Å. The contacts between proteins are very similar in all TRAF6 complexes and provide a molecular basis for understanding TRAF6 function.

**Architecture of the TRAF6 catalytic complex.** In the TRAF6-Ubc13~Ub complex structure Ubc13 binds at the canonical surface on the RING that includes loops 1 and 2[30], while the Ile36 patch of ubiquitin contacts Val108, Asp109 and Asn110 in the RING[26]. As well as interacting with the RING domain, ubiquitin contacts ZF1 and the connecting helix of the associated TRAF6 protomer (see below) (Fig. 1d, Supplementary Fig. 2e, f). In addition, the Ile44-centred hydrophobic patch on ubiquitin interacts with residues in α2 of Ubc13, and the conjugate adopts

the 'closed conformation'[26]. The same contacts are seen in all complexes suggesting that the Ubc13~Ub conjugate preferentially adopts the closed conformation when bound to TRAF6. Because the conjugate contacts both RINGs in the dimer, this structure accounts for the absolute dependence of ubiquitin chain assembly on TRAF RING dimerisation[12].

The C-terminal tail of ubiquitin has an extended conformation stabilised by Asp89 and Asp119 of Ubc13, but does not appear to be primed (Supplementary Fig. 2g). Two mutations (K92T and K94Q) were introduced to prevent non-specific ubiquitylation of Ubc13 and these probably account for the altered conformation of the tail of ubiquitin. However, the overall architecture of the TRAF6-Ubc13~Ub structure resembles the RNF4-Ubc13~Ub[26] and TRIM25-Ubc13~Ub[24] complexes and is compatible with the Ubc13~Ub/Mms2 complex[31] (Supplementary Fig. 3). This structure therefore provides a framework for understanding the transfer of ubiquitin onto Lys63 of another ubiquitin molecule, allowing the synthesis of Lys63-linked chains by TRAF6.

**Activation of the Ubc13~Ub thioester by ZF1 of TRAF6**. In the TRAF6-Ubc13~Ub complex, there are close contacts between ubiquitin and two Arg residues in the linker-helix and ZF1 of the interacting protomer. The sidechain of Arg126 from the linker-helix sits at the RING dimer interface and contacts Gly87 of the interacting RING domain, as well as the main chain of Lys33 at the base of the α-helix in ubiquitin. Whereas, Arg147 from ZF1 packs against Asp32 of ubiquitin (Fig. 2a, b). The density for both Arg residues was relatively well-defined (Supplementary Fig. 2b, f), and these residues are conserved in TRAF6 from a number of vertebrate species (Supplementary Fig. 4a).

To determine the importance of contacts between ubiquitin and the linker-helix and ZF1, we evaluated the activity of a series of TRAF6 mutant proteins. All mutants were made in $HsRZ_3$ and as a result the numbering of the residues mutated differs from the structure by one residue—we have attempted to make this clear by naming residues that were shown to be important (e.g. $Arg^{link}$ and $Arg^{ZF1}$). Single turnover assays where wild-type Ubc13 was first charged with Cy3-labelled ubiquitin containing a K63R mutation (*$Ub^{K63R}$) were used to assess activity. The purified conjugate was incubated with Uev1A and ubiquitin-D77 in the presence or absence of TRAF6 variants. Fluorescence imaging was used to monitor and quantify the appearance of diubiquitin (*$Ub^{K63R}$-Ub) over time (Fig. 2c). In these assays, wild-type $HsRZ_3$ efficiently promoted discharge of Ubc13 and diubiquitin formation, whereas the R125A ($Arg^{link}$) mutant appeared comparable to the no E3 control (Fig. 2c, d). Mutation of R146 ($Arg^{ZF1}$) also resulted in a substantial decrease in conjugate discharge, while mutation of Ser129 on the next turn of the linker-helix, and Glu144 did not impede activity. Similar results were obtained with multi-turnover chain-building assays (Fig. 2e).

While TRAF6 $RZ_3$ does not form a highly stable dimer[12] (Supplementary Fig. 4b), RING dimerisation is essential for TRAF6 E3 ligase activity. To determine if the Arg mutants influenced TRAF6 dimerisation, we analysed each variant by size exclusion chromatography (Supplementary Fig. 4c). All mutants were comparable to wild-type, except for the $Arg^{link}$ mutant which eluted slightly later, suggesting that the RING dimer had been destabilised. In the crystal structure $Arg^{link}$ contacts ubiquitin (Supplementary Fig. 2b, f), but it seems that it is also important for dimerisation. In this way $Arg^{link}$ resembles the aromatic residue at the C terminus of the dimeric RINGs of RNF4 and ML-IAP, which play dual roles in stabilising the RING dimer and the closed conformation of the conjugate[21, 32]. In contrast, mutation of $Arg^{ZF1}$ does not destabilise the dimer, but abrogates

activity, suggesting that this residue contacts ubiquitin and stabilises the closed conformation of the Ubc13~Ub conjugate. Together these data suggest that ZF1 is critical for the assembly of ubiquitin chains by TRAF6.

**TRAF6 dimers with a single E2~Ub binding site are active**. To further investigate the importance of ZF1 and RING dimerisation, we assessed the ability of inactive TRAF6 mutants to recover activity when mixed. In addition to the $Arg^{ZF1}$ mutant that prevents ubiquitin binding, we used a I72D/L74R (IL/DR) mutant of $HsRZ_3$ that does not interact with Ubc13 because the E2 binding interface is disrupted[12]. We hypothesised that each mutant would be inactive alone, but if the two TRAF6 RING species associate, the Ubc13-binding and ubiquitin-binding sites from different mutants could co-operate to form a complete binding site for the E2~Ub conjugate.

As expected the activity of the isolated mutants was significantly impaired in both the chain building and single turnover assays (Fig. 3). However, when the E2 interface mutant was mixed with the ZF1 mutant ($Arg^{ZF1}$) activity was considerably increased (Fig. 3). The increased activity must originate from a population of mixed TRAF6 dimers that have a single wild-type conjugate binding site comprising the E2 binding site on one RING, and ZF1 of the interacting $RZ_3$ molecule (as indicated in schematic in Fig. 3a, c). Together, these data suggest TRAF6 RING dimers that possess a single conjugate binding site retain activity. As a result, activity can be used to assess heterodimerisation of TRAF proteins.

**TRAF RING heterodimerisation**. Analysis of the sequences of the RING and ZF domains from the four most similar TRAFs (TRAF2, 3, 5 and 6) revealed an extended conserved region that comprises the RING dimer interface and the ubiquitin binding site on the linker-helix (Fig. 4a, b). The high level of similarity at the dimer interface, but relative absence of sequence conservation on the remainder of the surface, suggests that TRAF RING domain heterodimers may form. In addition, the Arg in the connecting helix is conserved in all four TRAFs, suggesting that other TRAFs might have the capacity to interact with ubiquitin.

To investigate the formation of TRAF RING heterodimers, we expressed $RZ_3$ from TRAF5 and analysed its ability to interact with TRAF6 using size-exclusion chromatography (Fig. 4c). TRAF5 and TRAF6 eluted together and earlier than either protein alone, suggesting that they preferentially form a stable dimer. In support of the increased stability of the heterodimer the mixture appeared to have greater activity than TRAF6 alone, even though TRAF5 had diminished activity with Ubc13-Uev1A in both single- and multi-turnover assays (Fig. 4d, left).

To confirm that TRAF5-TRAF6 heterodimers were active, we assessed the ability of TRAF5 to restore the activity of the $Arg^{link}$ and $Arg^{ZF1}$ TRAF6 mutants. Substantial discharge of Ubc13~Ub and recovery of chain-building activity was observed when TRAF5 $RZ_3$ was mixed with TRAF6 $Arg^{link}$ (Fig. 4d, middle), showing that TRAF5 has an intact ubiquitin binding site. This also shows that TRAF6 $Arg^{link}$ retains the ability to form dimers with TRAF5 even though the mutation disrupts TRAF6 homodimers (Supplementary Fig. 4c), which is consistent with the greater stability of the TRAF5-TRAF6 heterodimers. In contrast, when the Arg in the linker-helix of TRAF5 (Arg101) was mutated (R101A, referred to as $RZ_3$-$Arg^{link}$) activity was not recovered in an equivalent mixing experiment. This is probably because, like the equivalent TRAF6 mutant, the R101A mutation destabilises dimer formation (Supplementary Fig. 4d). Together, the data suggest that $Arg^{link}$ plays comparable roles in both TRAF5 and TRAF6.

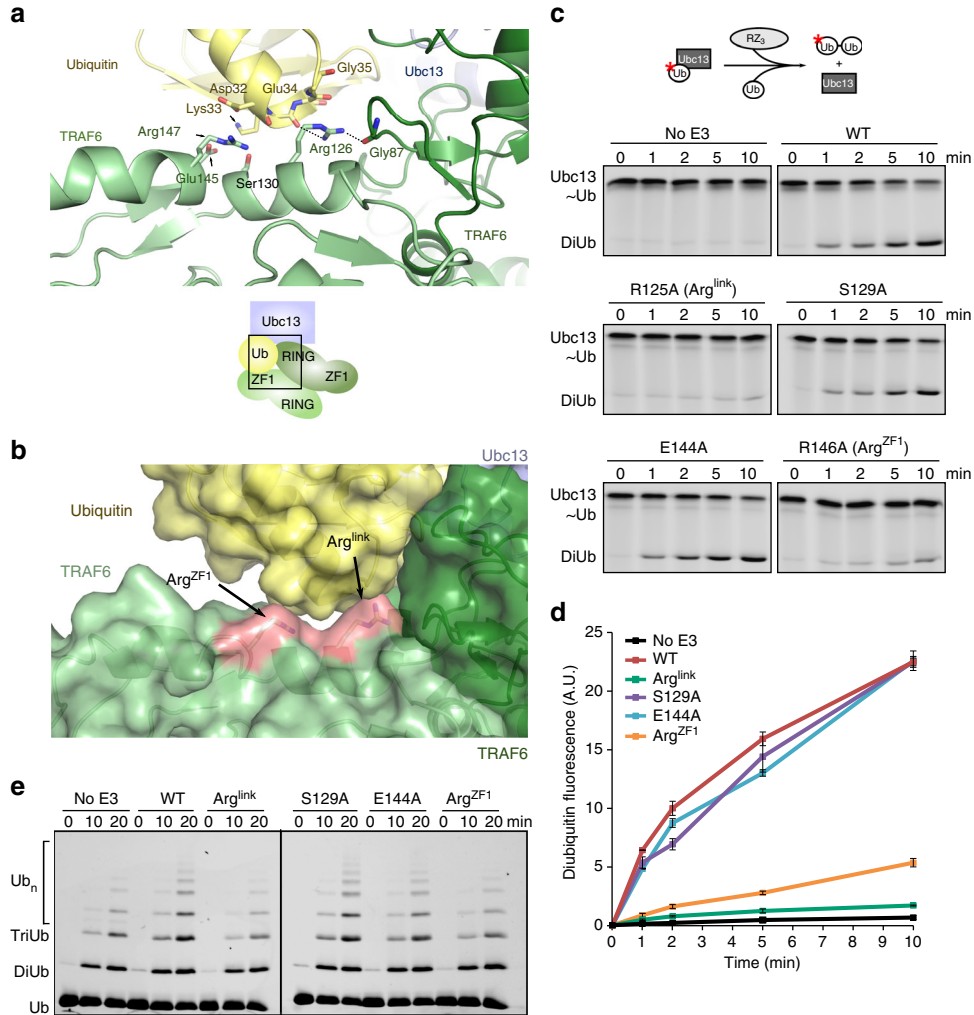

**Fig. 2** The linker-helix and ZF1 of TRAF6 stabilise ubiquitin so that it is primed for attack. **a** Details of the molecular contacts between the RING dimer (two shades of green) and ubiquitin from the RZ₁ complex (chains A, B and C). Predicted polar contacts are indicated with dashed lines. Carbon atoms coloured as in Fig. 1d, oxygen and nitrogen atoms coloured in red and blue, respectively. Numbering is as for *D. rerio* TRAF6. **b** Surface representation of the TRAF6–ubiquitin interaction. Critical residues for stabilisation are in pink. Arg126 is labelled as Arg$^{link}$; Arg147 as Arg$^{ZF1}$. **c** Ubc13 conjugated to fluorescently labelled ubiquitin was used in discharge assays in combination with no E3, wild-type *Hs*RZ₃, or mutant forms of TRAF6 as indicated. **d** Quantification of the formation of diubiquitin. Each experiment was performed in duplicate; error bars indicate range of measurement. **e** Chain-building assay using the same mutants as in panel **c**. Reaction mixture was spiked with 10% fluorescently labelled ubiquitin to allow visualisation of the chains

TRAF5 RZ₃ also recovered activity when mixed with the TRAF6 Arg$^{ZF1}$ mutant in both single- and multi-turnover assays (Fig. 4d, right). In TRAF5 Arg$^{ZF1}$ is replaced by a Gly, but the adjacent residue is an Arg (Fig. 4b). We reasoned that small adjustments in the structure of TRAF5 might allow this Arg residue to fulfil the role of Arg$^{ZF1}$ in TRAF6. In support of this, when TRAF5 in which Arg124 was mutated to Ala (R124A) was mixed with the TRAF6 Arg$^{ZF1}$ activity was not recovered to the same extent as observed when WT TRAF5 was included.

Together these data indicate that the linker-helix and ZF1 in TRAF5 are important for activity. In addition, these results provide evidence that TRAF RING heterodimers form and promote ubiquitin transfer.

## Discussion

TRAF proteins are adaptor molecules that have a trimeric 'TRAF' domain, which recognises the intracellular tails of membrane-embedded cytokine receptors, and also a dimeric RING domain that assembles ubiquitin chains in response to cytokine signalling.

Here we account for the importance of TRAF RING dimers and show that the ZF1 domain of TRAF6 interacts with ubiquitin to stabilise the closed conformation of the Ubc13~Ub conjugate. Thus, ZF1 in TRAF6 plays a critical role in accelerating K63 chain synthesis by Ubc13-Uev1A. We also demonstrate that the RING domains of TRAF5 and TRAF6 form a stable heterodimer that is an active E3 ligase, supporting the notion that TRAF hetero-dimers can regulate ubiquitylation.

The conformation of the Ubc13~Ub conjugate bound to TRAF6 resembles that reported for the RNF4 and TRIM25 complexes with Ubc13~Ub, and is compatible with the assembly of K63-linked ubiquitin chains by TRAF6[24, 26]. However, even though the RING-E2~Ub interactions are conserved (Fig. 5a), the RING-RING dimer interfaces vary, and accordingly, the ubiquitin and E2 moieties bound to the distal RING are rotated relative to the proximal molecules. As a result the critical cross-dimer ubi-quitin-stabilising interactions are shifted (Fig. 5a, b) so that in TRAF6 ubiquitin points towards the linker-helix and ZF1, where it interacts with basic residues (Arg$^{link}$ and Arg$^{ZF1}$). In this way, basic residues C-terminal to the TRAF6 RING domain appear to

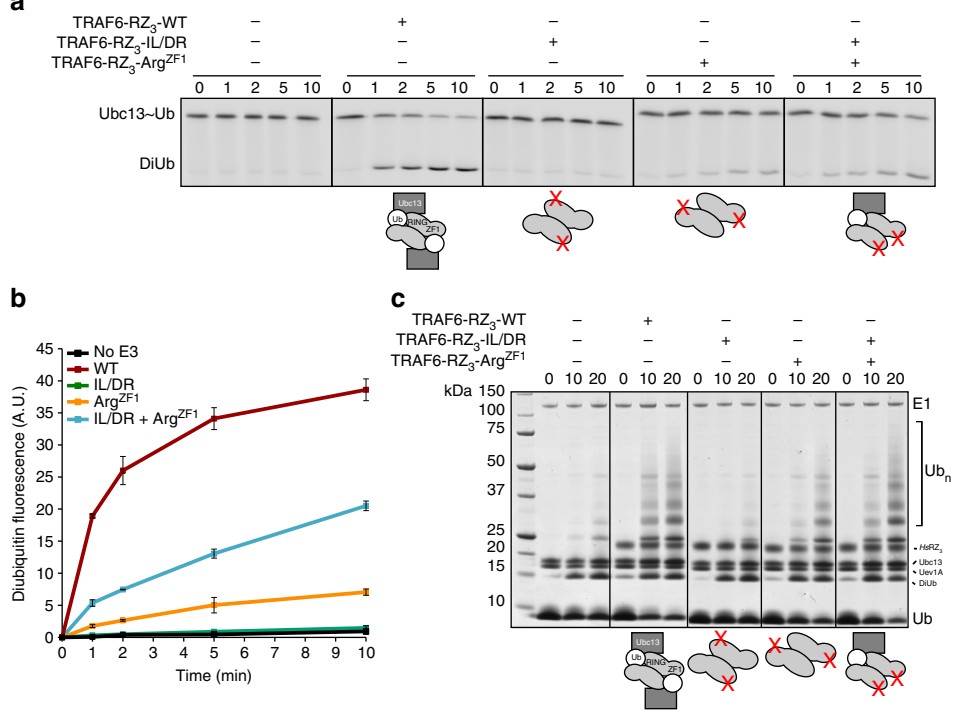

**Fig. 3** TRAF6 dimers with a single conjugate binding site are active. **a** Single-turnover assay using fluorescently labelled conjugate performed as in Fig. 2c. Mutant proteins that disrupted either the E2 interface (IL/DR) or ubiquitin binding (Arg$^{ZF1}$) were incubated with the conjugate alone, or following mixing (far right). Schematics indicate the site of each mutation on TRAF6. **b** The formation of diubiquitin from panel **a** was quantified. Experiments were performed in duplicate; error bars represent range of measurements. **c** Coomassie blue stained chain-building assay using TRAF6 RZ$_3$ mutants from panel **a**

fulfil the role of the well-established locking aromatic residue in other dimeric RINGs, such as RNF4 and ML-IAP[19, 21, 26].

A more general role for basic residues as the 'locking pin' seems likely because TRIM25 has similarly positioned residues, Lys65 and Thr67, which make comparable contacts with ubiquitin (Fig. 5b)[23, 24]. In fact, the RING domains of TRIM25 and TRAF6 overlay remarkably well, and this similarity extends beyond the RING domain to include the TRAF6 linker-helix and the α-helix that links the TRIM25 RING to its B-box domain. A helix C-terminal to the RING domain is also found in the structures of other TRIMs, and in RNF125 and LNX2 (Fig. 5c)[23, 33–35]. Each of these helices contains a basic residue in a position analogous to Arg$^{link}$ and, if dimers form, these residues could contact ubiquitin and stabilise the bound E2~Ub conjugate in the closed conformation.

TRAF proteins function downstream of a variety of inflammatory sensors, including tumour necrosis factor (TNF), Toll-, NOD- and RIG-I-like receptors[10, 36]. The specific signalling outputs from these receptors are determined by the complement of TRAF proteins recruited and the downstream effectors generated, in particular the ubiquitin chains which are built by TRAF proteins[2, 3, 10]. Even though Lys63-linked chains synthesised by TRAF6 have a critical role in signalling, the extent of ubiquitin chain synthesis by other TRAFs is uncertain. The E2-interacting surface of TRAFs is variable (Supplementary Fig. 5) and if E2s bind[16], it seems likely that each TRAF will bind to different E2s. However, formation of TRAF RING heterodimers, as demonstrated here, raises the possibility that even TRAFs that do not bind E2s have the potential to regulate ubiquitin transfer by providing a functional ZF1-linker-helix region to stabilise the catalytically primed complex that is recruited by a partner TRAF.

TRAF RING heterodimers have not been reported before, but multiple TRAFs are frequently found associated with signalling complexes[10, 36]. Furthermore, the coiled coil/TRAF-C domains of TRAF1 and TRAF2 are known to form a heterotrimer comprising one copy of TRAF1 and two copies of TRAF2[37]. TRAF1 does not possess a RING domain and these 1:2 TRAF1/2 heterodimers likely favour TRAF2 RING dimerisation. However, this raises the potential that other TRAF heterotrimers might form. Irrespective of whether heterotrimers form, there remains a crucial symmetry mismatch in that trimeric TRAFs interact with trimeric receptors, but RING domain dimers assemble ubiquitin chains. This symmetry mismatch is likely to be exacerbated in cells because many cytokine receptors signal via receptor clustering[38], where several TRAF trimers would be expected to come into close proximity. As a result, even if TRAF homotrimers form there is considerable potential for TRAF RING heterodimers to bridge trimeric TRAF complexes at the cell membrane. For example, in the simplest case it would be anticipated that if TRAF homotrimers form and a RING dimer is favoured, one RING domain would be free to interact with the RING of a neighbouring trimer as in Fig. 6. Alternatively, if RING heterodimers are favoured it is possible that two trimers might associate to form three RING heterodimers. Other combinations are also possible and may be favoured by membrane clustering as suggested previously[12, 39].

Regulation of TRAF proteins is complex, and much work remains to establish how trimeric TRAFs couple with dimeric RING domains to transduce signals. However, our data not only account for the importance of TRAF RING dimers, but also provide a mechanism by which clustering of cytokine receptors could elicit differential ubiquitin ligase activity—and the assembly of distinct ubiquitin chains from homo- and heterodimeric TRAF complexes. The nature and extent to which such signalling occurs in a cellular setting will depend on identification of the E2s bound

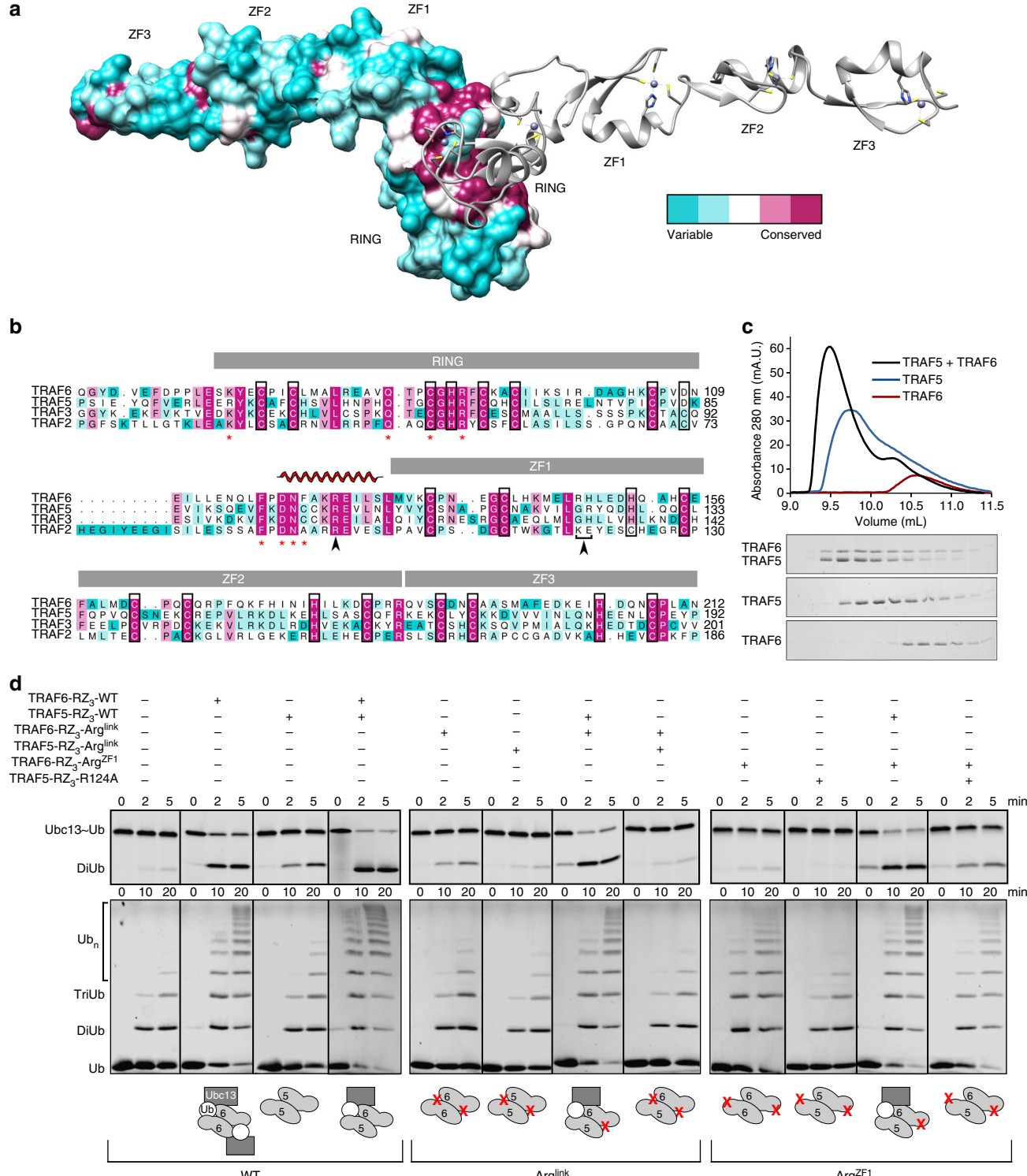

**Fig. 4** The TRAF RING dimer interface is conserved allowing formation of TRAF heterodimers. **a** Conservation surface mapping of the RING and first three ZFs of TRAFs 2, 3, 5 and 6 mapped on the structure of TRAF6 (PDB: 3HCS) using Chimera[51]. Magenta represents identical residues, while cyan shows low levels of similarity. The interacting RZ₃ molecule is shown in ribbon format (grey). Zinc atoms are shown as spheres, while coordinating side chains are shown as sticks with oxygen, nitrogen and sulphur atoms in red, blue and yellow, respectively. **b** Sequence alignment with identical colouring as on surface in panel **a**. Residues important for dimerisation are indicated by red asterisks, zinc-coordinating residues by black boxes, and the Arg residues at the ubiquitin binding interface are indicated by arrows. Domains are shown as grey boxes, and the linker-helix is indicated above the sequence. **c** Analytical size exclusion profile of TRAF6, TRAF5 and a mixture of the two. Equivalent fractions from each run were resolved by 16% SDS-PAGE and stained with Coomassie Blue. **d** Top: discharge of fluorescently labelled Ubc13~Ub thioester conjugate following addition of TRAF6, TRAF5, mutants of each, and mixtures of these proteins as indicated. Bottom: multi-turnover assays of the same TRAF proteins using fluorescently labelled ubiquitin

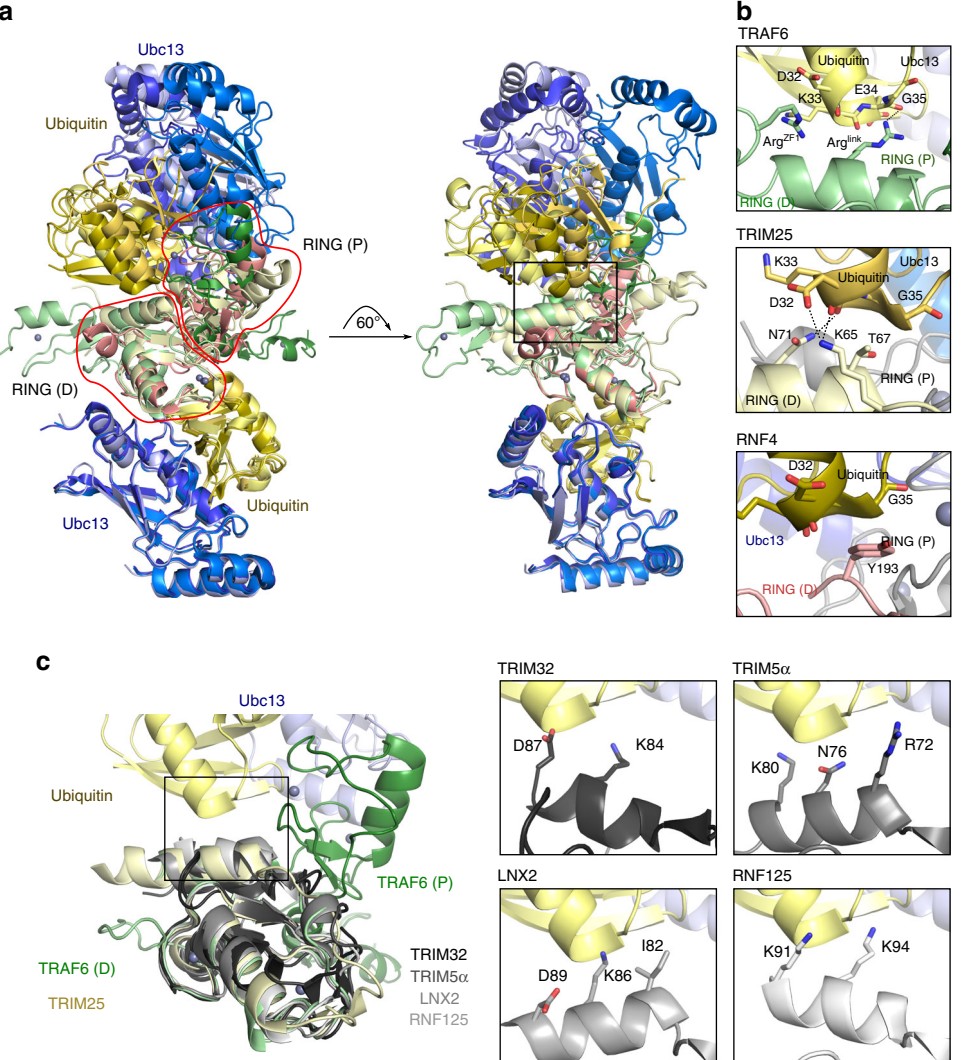

**Fig. 5** Conserved interactions stabilise the position of ubiquitin in RING-Ubc13-Ub complexes. **a** The distal (D) RING and bound E2~Ub conjugate from TRIM25-Ubc13-Ub (PDB: 5EYA) and RNF4-Ubc13-Ub (PDB: 5AIU) were overlaid onto TRAF6-Ubc13-Ub. Zinc atoms are shown as purple spheres. (D) refers to the distal RING, (P) proximal. The RING domains are indicated with red outlines. **b** A detailed view of the contacts between the distal RING and ubiquitin that stabilise the closed conformation. Predicted hydrogen bonds are indicated by dashed lines, and sticks are coloured by atom type (oxygen, nitrogen and sulphur atoms are coloured red, blue and yellow, respectively. Carbon atoms are coloured the same as ribbon). **c** An overlay of four RING domains (PDBs: TRIM5α 4TKP; TRIM32 5FEY; LNX2 5DIN; RNF125 5DKA) on the RING from *Dr*RZ$_3$ highlighting the conserved position of the C-terminal alpha helix. On the right is a detailed view of the putative ubiquitin-gripping residues from these RING domain structures. RNF125 is modelled as a dimer, though it is monomeric in PDB 5DKA

by other TRAFs, and thus the ubiquitin chains assembled, as well as systematic analysis of TRAF RING heterodimer formation. Despite these uncertainties, heterodimeric TRAF RING domains bridging a network of trimeric TRAF proteins is an attractive mechanism to account for the complexity of signals elicited downstream of many cytokine receptors.

## Methods

**Protein production and purification.** All proteins were expressed in *Escherichia coli* BL21 (DE3) (Novagen). Ubiquitin and ubiquitin variants were expressed as untagged proteins[40]. After overnight expression at 18 °C, the *E. coli* cells were harvested and re-suspended in 50 mM ammonium acetate pH 4.5 containing 1 mM EDTA. After sonication and clarification, the supernatant was injected on a 5 mL HiTrap SP column (GE) and eluted with a 50 mL linear gradient from 0 to 1 M NaCl, in 50 mM ammonium acetate pH 4.5, 1 mM EDTA. The peak fractions were pooled and injected on a HiLoad Superdex 75 16/600 column (GE) equilibrated in 20 mM Tris-HCl pH 7.5, 150 mM NaCl.

The human E1 protein was expressed with an N-terminal His$_6$ tag[41]. After lysis in 20 mM Tris-HCl pH 8.5, 100 mM NaCl, 5 mM imidazole, the supernatant was loaded on a 5 mL HisTrap column. A linear 50 mL gradient from 5 to 250 mM

imidazole was applied. The fractions containing E1 were injected on a HiLoad Superdex 200 16/600 column (GE) equilibrated with 20 mM Tris-HCl pH 7.5, 150 mM NaCl.

*Homo sapiens* TRAF6 constructs (*Hs*RZ$_3$: residues 50–211, 20.0 kDa; *Hs*RZ$_1$: 50–159, 13.8 kDa) were cloned into pET21d encoding a C-terminal His tag, while *Danio rerio* TRAF6 constructs (*Dr*RZ$_3$: residues 50–213, 19.7 kDa; *Dr*RZ$_1$: 50–159, 13.7 kDa) were in pET24b. All TRAF6 constructs were induced at an O.D.$_{600nm}$ of 0.6 with 0.2 mM IPTG and 0.1 mM ZnCl$_2$ and incubated at 28 °C for 4 h. Cells were sonicated in 50 mM Tris-HCl pH 7.5 and 350 mM NaCl, before being clarified and loaded into a 5 mL HisTrap FF column (GE). After elution, the peak fractions were loaded into a 10/300 Superdex 200 column (GE). TRAF5 was cloned into the pGEX6P3 vector containing an N-terminal GST tag. Protein expression was induced as for TRAF6, but was incubated at 18 °C overnight. Cells were sonicated in 20 mM Tris-HCl pH 7.5, 150 mM NaCl, and the lysate was clarified and bound to glutathione sepharose resin for 1 h. The GST tag was cleaved by overnight incubation with 3C protease. The soluble fraction was loaded on a 5 mL HiTrap Q column, eluted with a linear 1 M NaCl gradient, and cleaned up by size-exclusion chromatography as for TRAF6. Ubc13 and Uev1A were expressed as N-terminal GST-fusion proteins and purified using glutathione affinity, proteolysis of the GST tag, and size-exclusion chromatography.

For stable Ubc13~Ub conjugate formation, a mutated form of Ubc13 was used. This variant included C87K, K92T and K94Q mutations to allow formation of a

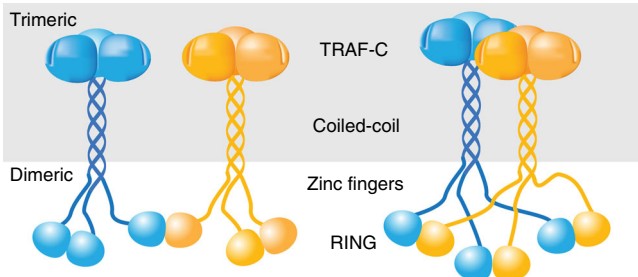

**Fig. 6** Model showing how TRAF trimers might associate to enable RING dimerisation. Two potential ways that TRAF RINGs can form homo- or heterodimers that may resolve the symmetry mismatch between the trimeric TRAF-C domain and the dimeric RING domain.

stable isopeptide-linked conjugate (Ubc13~Ub) while disrupting formation of other species as a result of non-specific transfer from the E1. For production of isopeptide-linked conjugates, 40 μM of pure Ubc13 (C87K, K92T, K94Q) was mixed with 80 μM wild-type ubiquitin, 1 μM E1, 1 mM ATP, and ATP cycling buffer containing 25 mM Tris-HCl pH 9.0, 5 mM MgCl₂, 10 mM phosphocreatine (Sigma) and 0.5 U mL⁻¹ creatine phosphokinase (Sigma)[42]. The reaction was incubated at 37 °C for 16 h. The resulting Ubc13~Ub conjugate was purified using a 26/600 Superdex 75 column (GE) equilibrated in 20 mM Tris-HCl pH 7.5, 150 mM NaCl.

**Pull-downs and analytical size-exclusion chromatography**. To assess binding of TRAF6 RZ₃ to Ubc13 and Ubc13~Ub conjugates, GST-fused Uev1A was mixed with TRAF6, Ubc13 and Ubc13~Ub conjugates. The incubations were performed in PBS containing 0.2% TWEEN 20 and 1 mM DTT, at 4 °C for 1 h. Subsequently, the resin was washed three times with PBS with 0.2% TWEEN 20 before being mixed with SDS loading dye and separated by SDS PAGE. The gels were stained with Coomassie Blue.

Analytical size-exclusion chromatography (SEC) was performed by injecting 200 μL of purified protein, or mixtures of proteins (DrRZ₃ and Ubc13~Ub at 250 μM; HsRZ₃ and Ubc13/Ubc13~Ub at 50 μM), over a 10/300 Superdex 200 (GE) equilibrated with 20 mM Tris-HCl pH 7.5, 150 mM NaCl at 0.5 mL min⁻¹. The recovered fractions were analysed by SDS PAGE. For the analysis of TRAF6 proteins, each protein was injected over a 10/300 Superdex 200 Increase at 40 μM. For determination of the oligomeric status of TRAF6, multiangle light scattering (MALS; Dawn 8+, Wyatt) coupled to a Superdex 200 10/300 GL (GE) was used. Each TRAF5 protein was analysed at 40 μM using a 10/300 Superdex 75 Increase column.

**Crystallisation and structure solution**. Purified *D. rerio* TRAF6 (RZ₁ and RZ₃) were mixed with isopeptide-linked Ubc13~Ub conjugate for 30 min on ice before being separated using size-exclusion chromatography over a 10/300 Superdex 200 column (GE). The fractions corresponding to the TRAF6-Ubc13~Ub complex were pooled, concentrated to ~2 mg mL⁻¹, and the PACT and JCSG+ crystal screens (Molecular Dimensions) were set up with a 1:1 ratio of protein to precipitant in sitting well plates (Swissci). Trays containing DrRZ₃ and DrRZ₁ resulted in crystals within 12 h in many conditions, which diffracted to ~8 Å. Crystals of DrRZ₃ were grown in 100–200 mM Na/K tartrate, 11–15% PEG 3350 and 100 mM Bis-Tris propane pH 7.5; DrRZ₁ crystals were produced in 0.05–0.3 mM sodium citrate, 100 mM Bis-Tris propane, and 17–23% PEG 3350. Fine tuning of the crystal conditions, changing drop ratios to 2:1 (protein: well solution), and slowing down the growth of crystals using microbatch resulted in DrRZ₁ and DrRZ₃ crystals that diffracted to 3.4 and 3.9 Å, respectively, on MX2 at the Australian Synchrotron[43].

Data sets were collected for each crystal form, and processed and scaled using XDS[44]. Data were merged with Aimless[45], and the structures were solved with Phaser-MR[46] using the complex of Ubc13-TRAF6 RING domain (PDB: 3HCT[12]) followed by ubiquitin (PDB: 1UBQ[47]). For one of the DrRZ₃ molecules, two of the three ZF could be automatically placed, while the third ZF had to be manually built using Coot[48]. Structures were refined using Refmac[49], while Coot was used to iteratively build missing residues, including the C-terminal tail of ubiquitin. For DrRZ₃, 92.1% of residues are in favoured regions, 6.6% in allowed regions and 1.3% are outliers. For DrRZ₁, 96.3% of residues are in favoured regions, 3.5% in allowed regions and 0.2% are outliers.

**Purification of fluorescently labelled conjugate**. Fluorescently tagged ubiquitin was produced by introducing a Cys residue before the beginning of the ubiquitin sequence, resulting in Met-Cys-ubiquitin. To label the free Cys purified ubiquitin was reduced with 1 mM TCEP and mixed with 5 μL of Cy3 for 1 h. The sample was desalted (5 mL HiTrap desalting, GE) to remove excess dye and incubated in the

dark at room temperature for ~16 h to allow unconjugated ubiquitin to form disulphide-linked dimers. Subsequently, the mixture was separated on a 16/600 HiLoad Superdex 75 column (GE) and the second peak corresponding to mono-meric ubiquitin was pooled and concentrated to ~2 mg mL⁻¹. Thioester-linked conjugate containing fluorescent ubiquitin was produced and purified using similar conditions as before,[50] except Ubc13 with a K97R mutation was mixed with ubiquitin, E1 and ATP cycling buffer at pH 7.5. The thioester charging reaction was incubated at 37 °C for 20 min before the conjugate was purified using a 10/300 Superdex 75 column (GE) equilibrated with 20 mM MES pH 6.5, 150 mM NaCl. The fractions containing conjugate were immediately flash frozen in liquid nitrogen.

**Ubiquitylation assays**. For multi-turnover ubiquitylation assays, 0.1 μM E1, 6 μM Ubc13 and Uev1A, 50 μM ubiquitin, and 4 μM TRAF proteins (wild-type or mutants) were mixed together in a buffer containing 20 mM Tris-HCl pH 7.5, 2 mM DTT, 2 mM ATP, 5 mM MgCl₂ and 0.1 M NaCl, and incubated at 37 °C. For fluorescently labelled multi-turnover assays, the reaction mixture was spiked with 10% fluorescent ubiquitin. At the indicated times, the reactions were quenched by adding SDS-PAGE loading buffer containing β-mercaptoethanol. Proteins were separated using 16% SDS-PAGE, 10–18% gradient SDS-PAGE (homemade) or commercial (4–12% bis-Tris SDS PAGE, Invitrogen) gels. Fluorescence was imaged using an Odyssey FC imaging system (LI-COR) at 600 nm with a 2 min exposure. Gels were stained with Coomassie blue.

For single-turnover discharge assays, 10 μM purified thioester linked Ubc13~Ub^K63R conjugate was mixed with 15 μM Uev1A, 50 μM D77 ubiquitin (a ubiquitin variant where the C-terminal Gly is mutated to Asp to prevent charging by E1), and 4 μM TRAF proteins and incubated at 20 or 37 °C for the indicated time. The final pH of the discharge reactions was 7.0. Reactions were quenched with non-reducing SDS-PAGE loading dye containing 10 mM N-ethylmaleimide. For quantification of fluorescently labelled conjugate, gels were scanned as for the multi-turnover assays and the intensity of fluorescent bands was quantified using Image Studio Lite (LI-COR).

**Data availability**. The coordinates and structure factors for TRAF6 RZ3 and RZ1 complexes with Ubc13~Ub have been deposited to the Protein Data Bank under the accession codes 5VO0 and 5VNZ, respectively. All other data are available from the corresponding authors upon reasonable request.

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

## Acknowledgements

We thank the New Zealand synchrotron group for facilitating access to the MX beamlines at the Australian Synchrotron, Victoria, Australia. A.D. acknowledges the support of a University of Otago Doctoral Scholarship. R.B. was supported by a Health Science Careers Development Award of the University of Otago. P.D.M. was supported by a Rutherford Discovery Fellowship from the Royal Society of New Zealand. This research was supported by funding from the Genesis Oncology Trust (NZ) and the Health Research Council of New Zealand.

## Author contributions

A.J.M. performed all structure determination and analysis. R.B. initiated the study. A.D., M.F. and J.Z. completed the assays and biochemical experiments. All authors analysed data. C.L.D. supervised this study and wrote the manuscript with A.J.M. and P.D.M.

## Additional information

**Competing interests:** The authors declare no competing financial interests.

