## [Peer Review file · Nature Communications]

Reviewers' comments:

Reviewer #1 (Remarks to the Author):

This short and focussed manuscript from Middleton, Budhidarmo and colleagues in the Day lab reports the first molecular rationale for the TRAF RING containing proteins using polymerisation (particularly dimerisation) as a strategy for activity. The analysis is restricted to the RING domains, and doesn't involve the coiled-coil repeats of TRAF6, which may contribute to oligomerisation. However, by crystallising the RING domain and first 3 zinc fingers of TRAF6, incomplete with an isopeptide linked Ubc13~Ub, the authors describe a mechanism by which TRAF6 RING domains dimerise to support binding of E2 on one RING and ubiquitin on the other. The mechanism is interesting and the authors provide data in support of the hypothesis that molecules deficient in either E2 or ubiquitin binding could rescue the opposite mutation.

The work presented is high quality, although the resolution of the structures is quite limited. I have a few comments and questions the authors may wish to consider when preparing a revised version of the manuscript.

Major comments: I may have missed something, but to me it looks in the structures as though both TRAF6 molecules bind both ubiquitins. This may be due to the orientation of the molecules in figure 1d, and without the pdb file to hand it is impossible for me to tell, but there seem to be contributions to ubiquitin binding from both RINGs. Perhaps this is the point, but I understood the authors' case to be that RING 1 binds E2, RING 2 binds ubiquitin.

All of the assays are carried out with Ubc13/Uev1a, but the crystal structure contains only Ubc13 isopeptide linked to ubiquitin. I appreciate the technical challenges in crystallising the full complex, but I think it would be useful if authors discuss how representative of the TRAF6-Ubc13-Uev1a-Ub-Ub complex they expect their structure to be.

The resolution could be mentioned in the introduction last paragraph. There is some confusion in the resolution - the table says 3.59, but the text refers to 3.9. I would also consider calling the resolution of the RZ1 structure 3.4 not 3.41. Also the intensity ($I/\sigma I$) for the RZ1 structure seems quite low at 1.2. Given the low resolution and the use of MR to solve the structures, some representative density of an omit map would be useful to help assess the quality of the structure.

In the results section I think RZ3 needs to be defined the first time it is introduced. Also for people not so familiar with all the TRAF proteins, an indication of the physical size of the fragments would be useful.

Do the TRAF RING domains dimerise in the absence of the E2~Ub conjugate? The data suggest that diubiquitin glues the TRAFs together, this is then extrapolated to suggest that this explains the need for dimerisation. How does the work fit with the higher order assemblies tested in Yin et al., 2009 (ref 12). Do the TRAF trimers influence the binding of Ubc13, or Ubc13~Ub, or ubc13-uev1a, or ubc13~ub-uev1a? Although some aspects of higher order oligomers are covered in the discussion, a figure outlining some possible models might help a reader navigate the current understanding.

p5 line 122 and line 128 'symmetrical' is misspelled.

p9 line 220 'moieties' is misspelled.

p10 line 249 comprised of should be comprising

p12 line 324, the references describing the pub files should be referenced - unless they are to contained in a paper, they are primary research findings.

Fig 1. It took me a while to see the no E3 and E144A traces - on my printer the two blues are very similar.

figure 3. I think calling the mixed mutants 'heterodimers' is misleading - heterodimers to me means two different proteins, not mutations of the same thing.

Reviewer #2 (Remarks to the Author):

This manuscript by Middleton and colleagues describes the E3 ubiquitin ligase activity of TRAF6. Disruption of TRAF6 function has been linked to cancer and immune signalling. TRAF6, in conjunction with its E2 conjugating partner Ubc13-UEV1 builds K63-linked polyubiquitin chains that are important in dictating the strength and duration of an immune response. The E3 ligase contains a C-terminal TRAF and coiled-coil domains needed for receptor binding. The N-terminus is marked by a RING domain, central for the ubiquitin process, and several Zn-finger domains including ZF1 that is necessary for activity. It is well established that TRAF6 must form a dimeric RING complex for full ubiquitination activity. This work aims to identify how the RING and ZF1 domains work together to propagate ubiquitin chain formation. To do this the authors determined the three-dimensional structure of RING+ZF1-3 from Danio Rerio TRAF6 (DrRZ3) bound to the Ubc13~Ub conjugate. An additional, a slightly higher resolution structure containing only RING+ZF1 was also obtained. The structures show that TRAF6 recruits the closed conformation of Ubc13~Ub by a conventional interaction between one RING protomer with the E2 enzyme. The conjugated Ub interacts with the RING and ZF1 (and its adjacent helix) domains in the adjacent TRAF6 protomer in such a way that a Ubc13~Ub complex bridges the two TRAF6 molecules in the dimer. The closed conformation of Ubc13~Ub appears to be stabilized by a pair of Arg residues in the ZF1 and helix and substitution of these severely hampers Ub transfer in a single turnover assay. The structure and accompanying ubiquitination assays account for TRAF6 dimerization to complete ubiquitin chain building activity and presents some unique aspects of Ubc13 ubiquitination that differs from other E2~Ub/E3 structures. This manuscript should have broad appeal to the ubiquitination community.

The authors also present some intriguing data regarding potential TRAF5/6 heterodimerization based on structure and sequence observations. The possibility of TRAF heterodimer formation is very important and would present a potential mechanism for signal modulation. In the structure Arg-ZF1 is not conserved between TRAF5/6 and substitution of this in TRAF6 has a significant impact in single and multiple turnover assays. Yet the heterodimer assays done in Fig. 4 with TRAF5/6 comprise both proteins deficient in a Ub binding required residue (TRAF6 - Arg-link, TRAF5-WT because it is a Gly residue). The premise of this experiment is that TRAF5 does not bind Ubc13. The authors should show experiments that show this (similar to Fig. 1). Even if this were true the heterodimer experiment (Fig. 4 lanes 16-18) would be expected to be less active due to an incomplete Ub binding site in TRAF5 (Gly residue at Arg-link site). Further, if all energetics between homo- and heterodimers were equivalent one would expect a maximum 50% of activity for a TRAF5/6 heterodimer compared to WT TRAF6. The data in Fig. 4 appears to show nearly 100% recovery so there is something else going on here. Despite its potential importance I feel the authors need to present a more compelling case to account to substantiate and clarify the role of heterodimerization.

Minor Comments

P3 (lines 63,64) – It is stated that the RING domain binds the E2 and determines the type of ubiquitin chains. It is true TRAF6 builds K63-linked chains with Ubc13/UEV1. If other types of chains can be built by TRAF6 (or other TRAF ligases) it would be best to indicate explicitly what these chains are and which TRAF proteins are involved. Otherwise such a general statement is rather misleading.

Have TRAF5/6 (or other TRAF proteins) been shown to heterodimerize by other methods?

A discussion/model of the Ubc13~Ub/TRAF6 ubiquitination that includes UEV1 would be useful as it relates to the current structure and mechanism.

Reviewer #3 (Remarks to the Author):

In the manuscript titled "The activity of TRAF RING homo- and heterodimers is regulated by Zinc Finger 1" by Middleton et al., the authors solved the structures of the dimeric *Drosophila* TRAF6 RING E3 ubiquitin ligase in complex with an E2-ubiquitin (Ubc13-Ub) conjugate. In conjunction with biochemical data the authors show why TRAF6 dimerization is necessary for its activity: While the RING domain of one TRAF6 protomer binds the E2, residues in the ZF1 domain and the linker between the RING and ZF1 domain from the other TRAF6 protomer bind ubiquitin to stabilize the closed E2-ubiquitin conformation necessary for ubiquitin transfer. The authors also present biochemical data that suggests that TRAF6/TRAF5 heterodimers can transfer ubiquitin in a similar manner. Finally, they hypothesize that the observed interactions between ubiquitin and the linker helix may be a more general structural feature in other dimeric RING E3 ligases to stabilize the closed conformation of the E2-ubiquitin conjugate.

While the structure for the first time shows how TRAF6 binds the E2-ubiquitin conjugate and why TRAF6 dimerization is necessary for its E3 ligase activity, I am not fully convinced by some of the conclusions and that these observations are transferable to other TRAF homodimers and to TRAF heterodimers. However, if the authors can address my concerns as detailed below, I recommend publication in Nature Communications.

Specifically, the following points need to be addressed:

The authors identify two Arg residues (named Arg-ZF1 and Arg-link) that in their structure interact with ubiquitin and that are important for TRAF6 activity in E2 discharge assays and chain-building assays. Arg-link is conserved in TRAF6 of several vertebrate species (Fig S3) and other TRAF proteins (Fig 4b). However, additional to its interaction with ubiquitin as observed by the authors, it also interacts with residues from the other TRAF protomer (as illustrated in Fig 2a of the current manuscript and also pointed out in Fig 1f of ref 12 [Yin 2009, NSMB]). Thus, the observed loss of function may not be due to the interaction with ubiquitin but also because TRAF dimerization is impaired. The authors need to verify that the TRAF6 Arg-link mutant can still form dimers, for example using size-exclusion chromatography or analytical ultracentrifugation. Then the authors should test the mutant of this conserved residue in the TRAF6 mixing studies (Fig 3) instead of just the less conserved (see below) Arg-link mutant.

The second important residue identified by the authors (Arg-ZF1) is only conserved among TRAF6 in different species, but not in other TRAF proteins (Fig 4b and S3). This questions the general importance for other TRAF proteins of the interactions between ubiquitin and ZF1 as claimed by the authors. In particular, in the assay showing activity of TRAF5/6 heterodimers (Fig 4), the TRAF5 protein could be seen as a Arg-ZF1 mutant since it does not have a Gly residue instead of the Arg-ZF1 residue (Fig 4b). In light of this and the claim by the authors that Arg-ZF1 is important for ubiquitin binding and TRAF activity, how do the authors explain that the TRAF5/6 heterodimers are still active and how their observations in TRAF6 are relevant for other TRAF proteins?

Furthermore, what is the reason to use the TRAF6 Arg-link mutant in the TRAF5/6 assay instead of the TRAF6 Arg-ZF mutant used for the same type of assay in Fig 3? My general concern here goes in the same direction as above and is rooted in the fact that Arg-link might be relevant for TRAF dimerization. The TRAF5/TRAF6 Arg-link heterodimer might be more active than the TRAF6 Arg-link homodimer, because in the heterodimer only one half of the dimer interface is mutated

(TRAF6 but not TRAF5), whereas in the TRAF6 Arg-link homodimer both are. The authors need to perform this assay using the TRAF6 Arg-ZF1 mutant additional to the Arg-link mutant used.

Concerning the TRAF5/6 heterodimer assay, the authors claim that TRAF5 does not bind Ubc13 (p. 8, l. 194). However, their data does not show this since lack of ubiquitin chain building could have numerous other reasons besides this. The authors should either remove this statement or support it by relevant data or references to literature.

One other important point is that it is unclear how often the E2-ubiquitin discharge assays were performed. The legend to Fig 2d states that "each measurement was performed in duplicate", whereas in a similar experiment in Fig 3b, it states that "experiments were performed in duplicate". Does this mean that for Fig 2d, a single gel was quantified (measured) twice or was the full experiment performed twice? Please clarify this and perform a second independent experiment if this has not been done.

Minor points:

Fig 2a: The position of some of the labels is confusing, especially around Arg147.

Fig 5c: The different colors are not defined. While green and yellow might be clear from previous figures, the light yellow and different shades of grey (?) RING domains are not clear, especially since the colors in the smaller panels on the right seem to be different. In general, the left part of the figure is very busy and it is hard to focus on the key point, i.e. that the RING domains and C-terminal helices of the different RING proteins superimpose. It may be a good idea to have another panel that just shows the RING domains and helices (and maybe the ubiquitin) without the non-superimposing domains in the background.

Based on the descriptions in the manuscript, it is unclear if in the structures the ubiquitin C-terminus is in the primed state or not. While in the main text (p. 6, l.136) the authors state that the C-terminus "does not appear to be primed", the figure legend for Figure S2f says: "Detailed interaction between the primed C-terminal tail of ubiquitin and Ubc13". Please clarify this.

Supplementary Figure 2c: The authors have to define what type of electron density map is shown and at what contour level.

Fig 5b: The authors should label the ubiquitin residues in the right and middle panels.

Some of the color choices in the figures are not optimal when printed. Examples are:

- Fig 1a: Black on green is difficult to read
- Fig 2d: The two shades of blue indicating no E3 and the E144A mutant very similar and should be changed

The authors should carefully check the manuscript for typos and errors. Some examples are:

- p. 5 l. 122 'symmetrical' should be symmetrical.
- p. 59 l. 222: 'Fig. S5a,b' should be Fig. 5a,b.
- Fig 5c: In the TRIM5a panel K80 should be R80 and R72 should be K72.
- Table S1: The actual PDB codes should be given in the header column instead of the words "PDB code".

Reviewer #1 (Remarks to the Author):

This short and focussed manuscript from Middleton, Budhidarmo and colleagues in the Day lab reports the first molecular rationale for the TRAF RING containing proteins using polymerisation (particularly dimerisation) as a strategy for activity. The analysis is restricted to the RING domains, and doesn't involve the coiled-coil repeats of TRAF6, which may contribute to oligomerisation. However, by crystallising the RING domain and first 3 zinc fingers of TRAF6, incomplete with an isopeptide linked Ubc13~Ub, the authors describe a mechanism by which TRAF6 RING domains dimerise to support binding of E2 on one RING and ubiquitin on the other. The mechanism is interesting and the authors provide data in support of the hypothesis that molecules deficient in either E2 or ubiquitin binding could rescue the opposite mutation.

The work presented is high quality, although the resolution of the structures is quite limited. I have a few comments and questions the authors may wish to consider when preparing a revised version of the manuscript.

Major comments: I may have missed something, but to me it looks in the structures as though both TRAF6 molecules bind both ubiquitins. This may be due to the orientation of the molecules in figure 1d, and without the pdb file to hand it is impossible for me to tell, but there seem to be contributions to ubiquitin binding from both RINGs. Perhaps this is the point, but I understood the authors' case to be that RING 1 binds E2, RING 2 binds ubiquitin.

We have revised the description of the structure on pages 5 and 6 in an effort to make this clearer. In essence ubiquitin contacts the RING to which the E2 binds (as observed in other comparable structures), while ZF1 and the linker helix from the interacting molecule also contact ubiquitin.

All of the assays are carried out with Ubc13/Uev1a, but the crystal structure contains only Ubc13 isopeptide linked to ubiquitin. I appreciate the technical challenges in crystallising the full complex, but I think it would be useful if authors discuss how representative of the TRAF6-Ubc13-Uev1a-Ub-Ub complex they expect their structure to be.

The reviewer makes a good point that we have addressed by overlaying the Ubc13 in our RZ₃ crystal structure with the structure of Ubc13~Ub-Mms2 reported by Eddins *et al.* 2006 (Uev1A and Mms2 are almost identical and play the same role in positioning the acceptor ubiquitin). In this overlay, Mms2 does not clash with any molecules in the asymmetric unit, and the acceptor ubiquitin molecule from the Eddins structure is placed so that Lys63 is pointing towards the active site of the Ubc13 from our structure. Our structure is compatible with the K63-building complex of Ubc13 and Mms2 (or Uev1a).

We have included the overlay figure as Supplementary figure 3 and have commented in the text on page 6 - Architecture of the catalytic complex section.

The resolution could be mentioned in the introduction last paragraph. There is some confusion in the resolution - the table says 3.59, but the text refers to 3.9. I would also consider calling the resolution of the RZ1 structure 3.4 not 3.41. Also the intensity ($I/\sigma I$) for the RZ1 structure seems quite low at 1.2. Given the low resolution and the use of MR to solve the structures, some representative density of an omit map would be useful to help assess the quality of the structure.

We have corrected the typo, and rounded the resolution to 3.4 Å. While it is true that an $I/\sigma I$ of 1.2 is low, the CC1/2 value is >0.5 , which is generally regarded as acceptable. As a result, we feel that altering the cut-off will result in loss of useful information. We have added a representative omit map for the RZ₃ structure as *Supplementary Fig 2b*.

In the results section I think RZ3 needs to be defined the first time it is introduced. Also for people not so familiar with all the TRAF proteins, an indication of the physical size of the fragments would be useful.

We have now explicitly defined RZ₃ at the beginning of the results section, and included the mass of the different RING proteins in the Experimental Procedures section.

Do the TRAF RING domains dimerise in the absence of the E2~Ub conjugate? The data suggest that diubiquitin glues the TRAFs together, this is then extrapolated to suggest that this explains the need for dimerisation. How does the work fit with the higher order assemblies tested in Yin et al., 2009 (ref 12). Do the TRAF trimers influence the binding of Ubc13, or Ubc13~Ub, or ubc13-uev1a, or ubc13~ub-uev1a? Although some aspects of higher order oligomers are covered in the discussion, a figure outlining some possible models might help a reader navigate the current understanding.

The TRAF6 RINGs dimerise in the absence of the conjugate, although the dimer is not highly stable and samples typically contain a mixture of monomer and dimer. We have now included MALS data as Supplementary figure 4b). The structure and properties of the dimeric RING domains of TRAF6 are discussed in some detail by Yin et al and we have now referred to this more explicitly (Page 5). We fully expect that TRAF trimers (mediated by the coiled coil domains) might enhance RING dimerisation. We have elaborated this point in the discussion and included an additional schematic (Figure 6) to help make this point clearer. Further studies with longer proteins, ideally full-length will be required to resolve this point.

p5 line 122 and line 128 'symmetrical' is misspelled.

p9 line 220 'moieties' is misspelled.

p10 line 249 comprised of should be comprising

p12 line 324, the references describing the pub files should be referenced - unless they are to contained in a paper, they are primary research findings.

Fig 1. It took me a while to see the no E3 and E144A traces - on my printer the two blues are very similar.

The typos have been corrected, the missing references added, and the colouring on Fig. 1 has been adjusted.

figure 3. I think calling the mixed mutants 'heterodimers' is misleading - heterodimers to me means two different proteins, not mutations of the same thing.

We have revised this text and now refer to mixtures.

Reviewer #2 (Remarks to the Author):

This manuscript by Middleton and colleagues describes the E3 ubiquitin ligase activity of TRAF6. Disruption of TRAF6 function has been linked to cancer and immune signalling. TRAF6, in conjunction with its E2 conjugating partner Ubc13-UEV1 builds K63-linked polyubiquitin chains that are important in dictating the strength and duration of an immune response. The E3 ligase contains a C-terminal TRAF and coiled-coil domains needed for receptor binding. The N-terminus is marked by a RING domain, central for the ubiquitin process, and several Zn-finger domains including ZF1 that is necessary for activity. It is well established that TRAF6 must form a dimeric RING complex for full ubiquitination activity. This work aims to identify how the RING and ZF1

domains work together to propagate ubiquitin chain formation. To do this the authors determined the three-dimensional structure of RING+ZF1-3 from Danio Rerio TRAF6 (DrRZ3) bound to the Ubc13~Ub conjugate. An additional, a slightly higher resolution structure containing only RING+ZF1 was also obtained. The structures show that TRAF6 recruits the closed conformation of Ubc13~Ub by a conventional interaction between one RING protomer with the E2 enzyme. The conjugated Ub interacts with the RING and ZF1 (and its adjacent helix) domains in the adjacent TRAF6 protomer in such a way that a Ubc13~Ub complex bridges the two TRAF6 molecules in the dimer. The closed conformation of Ubc13~Ub appears to be stabilized by a pair of Arg residues in the ZF1 and helix and substitution of these severely hampers Ub transfer in a single turnover assay. The structure and accompanying ubiquitination assays account for TRAF6 dimerization to complete ubiquitin chain building activity and presents some unique aspects of Ubc13 ubiquitination that differs from other E2~Ub/E3 structures. This manuscript should have broad appeal to the ubiquitination community.

We thank the reviewer for their positive comments.

The authors also present some intriguing data regarding potential TRAF5/6 heterodimerization based on structure and sequence observations. The possibility of TRAF heterodimer formation is very important and would present a potential mechanism for signal modulation. In the structure Arg-ZF1 is not conserved between TRAF5/6 and substitution of this in TRAF6 has a significant impact in single and multiple turnover assays. Yet the heterodimer assays done in Fig. 4 with TRAF5/6 comprise both proteins deficient in a Ub binding required residue (TRAF6 – Arg-link, TRAF5-WT because it is a Gly residue). The premise of this experiment is that TRAF5 does not bind Ubc13. The authors should show experiments that show this (similar to Fig. 1). Even if this were true the heterodimer experiment (Fig. 4 lanes 16-18) would be expected to be less active due to an incomplete Ub binding site in TRAF5 (Gly residue at Arg-link site). Further, if all energetics between homo- and heterodimers were equivalent one would expect a maximum 50% of activity for a TRAF5/6 heterodimer compared to WT TRAF6. The data in Fig. 4 appears to show nearly 100% recovery so there is something else going on here. Despite its potential importance I feel the authors need to present a more compelling case to account to substantiate and clarify the role of heterodimerization.

The reviewer makes several good points regarding figure 4. As a result, we have revised this figure and the associated text (page 9).

- i. We have clarified the text regarding TRAF5 activity with Ubc13 to indicate that this is diminished, and not non-existent, compared to that of TRAF6. We thank the referee for drawing our attention to this.
- ii. We have expanded our discussion regarding the activity of the TRAF mixtures and analysed the TRAF5/6 mixture by SEC (Figure 4c). This shows that the TRAF5 and TRAF6 RZ₃ proteins preferentially associate to form a heterodimer. We also now include activity data for the WT TRAF5/6 mix showing this is increased compared to TRAF6 alone. Thus the apparent complete recovery of activity when we mix WT-TRAF5 and TRAF6-Arg mutants, even though only half the conjugate binding sites are expected if all molecules form TRAF5/6 heterodimers, likely relates to the increased stability of the TRAF5/TRAF6 RING dimer.
- iii. Lastly, we have extended this set of mixing experiments to include both TRAF6 mutants that disrupt activity. As shown in figure 4d, TRAF5 can complement the activity of both TRAF6 Arg^{link} and Arg^{ZF1}. In addition, we show that, although Arg^{ZF1} is not conserved in TRAF5, it appears that the adjacent Arg, in part, fulfils this role in TRAF5 because the

R124A mutant of TRAF5 does not recover activity as effectively as WT TRAF5. We have expanded the text to clarify these points.

Together these experiments significantly strengthen the heterodimer section of our manuscript and we thank the referees.

Minor Comments

P3 (lines 63,64) – It is stated that the RING domain binds the E2 and determines the type of ubiquitin chains. It is true TRAF6 builds K63-linked chains with Ubc13/UEV1. If other types of chains can be built by TRAF6 (or other TRAF ligases) it would be best to indicate explicitly what these chains are and which TRAF proteins are involved. Otherwise such a general statement is rather misleading.

We have modified the text to more correctly reflect the role of TRAF6.

Have TRAF5/6 (or other TRAF proteins) been shown to heterodimerize by other methods?

As far as we are aware, there are no reports of RING heterodimers for any TRAF proteins. Combinations of TRAFs have been reported to associate, but we are not aware of any experiments that would have revealed RING heterodimerisation. We hope that our study will provide others with the tools to more directly investigate the importance of TRAF RING heterodimerisation.

A discussion/model of the Ubc13~Ub/TRAF6 ubiquitination that includes UEV1 would be useful as it relates to the current structure and mechanism.

We have now included Supplementary figure 3 and have commented in the text on page 6 – Architecture of the catalytic complex section. (see response to Reviewer 1).

Reviewer #3 (Remarks to the Author):

In the manuscript titled “The activity of TRAF RING homo- and heterodimers is regulated by Zinc Finger 1” by Middleton et al., the authors solved the structures of the dimeric *Drosophila* TRAF6 RING E3 ubiquitin ligase in complex with an E2-ubiquitin (Ubc13-Ub) conjugate. In conjunction with biochemical data the authors show why TRAF6 dimerization is necessary for its activity: While the RING domain of one TRAF6 protomer binds the E2, residues in the ZF1 domain and the linker between the RING and ZF1 domain from the other TRAF6 protomer bind ubiquitin to stabilize the closed E2-ubiquitin conformation necessary for ubiquitin transfer. The authors also present biochemical data that suggests that TRAF6/TRAF5 heterodimers can transfer ubiquitin in a similar manner. Finally, they hypothesize that the observed interactions between ubiquitin and the linker helix may be a more general structural feature in other dimeric RING E3 ligases to stabilize the closed conformation of the E2-ubiquitin conjugate.

While the structure for the first time shows how TRAF6 binds the E2-ubiquitin conjugate and why TRAF6 dimerization is necessary for its E3 ligase activity, I am not fully convinced by some of the conclusions and that these observations are transferable to other TRAF homodimers and to TRAF heterodimers. However, if the authors can address my concerns as detailed below, I recommend publication in Nature Communications.

Specifically, the following points need to be addressed:

The authors identify two Arg residues (named Arg-ZF1 and Arg-link) that in their structure interact with ubiquitin and that are important TRAF6 activity in E2 discharge assays and chain-building assays. Arg-link is conserved in TRAF6 of several vertebrate species (Fig S3) and other TRAF proteins (Fig 4b). However, additional to its interaction with ubiquitin as observed by the authors, it also interacts with residues from the other TRAF protomer (as illustrated in Fig 2a of the current manuscript and also pointed out in Fig 1f of ref 12 [Yin 2009, NSMB]). Thus, the observed loss of function may not be due to the interaction with ubiquitin but also because TRAF dimerization is impaired. The authors need to verify that the TRAF6 Arg-link mutant can still form dimers, for example using size-exclusion chromatography or analytical ultracentrifugation. Then the authors should test the mutant of this conserved residue in the TRAF6 mixing studies (Fig 3) instead of just the less conserved (see below) Arg-link mutant.

The referee makes a good point regarding the contribution of Arg-link to dimerisation. We had overlooked this point in our original submission because when mixed with TRAF5, activity could be recovered indicating that TRAF6 Arg-link mutant could form heterodimers with TRAF5 (Fig. 4d, middle).

Since submitting our manuscript we have extended our analysis and analysed the TRAF6 mutants by SEC. The traces are now included as Supplementary Fig. 4c. The Arg-link mutant elutes slightly later than the WT suggesting that the dimer has been destabilised. We have not included the additional TRAF6 Arg-link mixing experiment in Figure 3 because activity is not recovered with the E2-interface mutant. However, we are happy to include this figure if that is preferred.

The inability to recover activity with the Arg-link mix is likely attributable to the instability of the TRAF6 dimer, and the importance of this residue for stabilising dimerisation. In contrast, the TRAF5/6 heterodimer appears to be considerably more stable and loss of this contact is tolerated, thus activity is recovered when TRAF5 and the TRAF6-Arglink mutant are mixed (Fig 4d, middle). This indicates that the Arg-link mutant is folded.

We have expanded the text to more correctly account for the role of Arg-link in TRAF6. Furthermore, we highlight the similarities to the aromatic residues at the RING dimer interface of some IAPs and RNF4. Noting that mutation of these aromatic residues also destabilises the RING dimer. We now emphasise the dual role of Arg-link, which suggests it is functionally similar to the C-terminal aromatic of ML-IAP and RNF4.

The second important residue identified by the authors (Arg-ZF1) is only conserved among TRAF6 in different species, but not in other TRAF proteins (Fig 4b and S3). This questions the general importance for other TRAF proteins of the interactions between ubiquitin and ZF1 as claimed by the authors. In particular, in the assay showing activity of TRAF5/6 heterodimers (Fig 4), the TRAF5 protein could be seen as a Arg-ZF1 mutant since it does have a Gly residue instead of the Arg-ZF1 residue (Fig 4b). In light of this and the claim by the authors that Arg-ZF1 is important for ubiquitin binding and TRAF activity, how do the authors explain that the TRAF5/6 heterodimers are still active and how their observations in TRAF6 are relevant for other TRAF proteins?

Furthermore, what is the reason to use the TRAF6 Arg-link mutant in the TRAF5/6 assay instead of the TRAF6 Arg-ZF mutant used for the same type of assay in Fig 3? My general concern here goes in the same direction as above and is rooted in the fact that Arg-link might be relevant for TRAF dimerization. The TRAF5/TRAF6 Arg-link heterodimer might be more active than the TRAF6 Arg-link homodimer, because in the heterodimer only one half of the dimer interface is mutated (TRAF6 but not TRAF5), whereas in the TRAF6 Arg-link homodimer both are. The authors need to perform this assay using the TRAF6 Arg-ZF1 mutant additional to the Arg-link mutant used.

The reviewer makes a number of important points regarding Figure 4. As discussed in response to reviewer 2 we have extended our analysis of the TRAF5/6 heterodimer and now show that (i) the RINGs of TRAF5 and TRAF6 preferentially associate by size exclusion chromatography (Fig 4c), and (ii) that TRAF5 can recover the activity of both TRAF6 Arg mutants (Fig. 4d, middle & right). We also show that although the Arg^{ZF1} residue is not conserved in TRAF5 the adjacent basic residue appears to fulfil a similar role. These additional experiments significantly strengthen the importance of ZF1 in TRAFs and provide direct evidence to suggest that TRAF5/TRAF6 heterodimers form.

Concerning the TRAF5/6 heterodimer assay, the authors claim that TRAF5 does not bind Ubc13 (p. 8, l. 194). However, their data does not show this since lack of ubiquitin chain building could have numerous other reasons besides this. The authors should either remove this statement or support it by relevant data or references to literature.

As the referee suggests we have revised this section and removed the comment about binding (see also the response to reviewer 2).

One other important point is that it is unclear how often the E2-ubiquitin discharge assays were performed. The legend to Fig 2d states that “each measurement was performed in duplicate”, whereas in a similar experiment in Fig 3b, it states that “experiments were performed in duplicate”. Does this mean that for Fig 2d, a single gel was quantified (measured) twice or was the full experiment performed twice? Please clarify this and perform a second independent experiment if this has not been done.

Both of these sets of experiments were performed in duplicate (technical duplicates). We have clarified this in our legend for Figure 2.

Minor points:

Fig 2a: The position of some of the labels is confusing, especially around Arg147.

The labels on the figure have been clarified.

Fig 5c: The different colors are not defined. While green and yellow might be clear from previous figures, the light yellow and different shades of grey (?) RING domains are not clear, especially since the colors in the smaller panels on the right seem to be different. In general, the left part of the figure is very busy and it is hard to focus on the key point, i.e. that the RING domains and C-terminal helices of the different RING proteins superimpose. It may be a good idea to have another panel that just shows the RING domains and helices (and maybe the ubiquitin) without the non-superimposing domains in the background.

The colours are now explicitly described on Fig 5c. We have removed the non-superimposing parts of the RING overlay to highlight the overlaid alpha helices. In addition, we have matched the colours between the panels on the right and the overlay on the left.

Based on the descriptions in the manuscript, it is unclear if in the structures the ubiquitin C-terminus is in the primed state or not. While in the main text (p. 6, l.136) the authors state that the C-terminus “does not appear to be primed”, the figure legend for Figure S2f says: “Detailed interaction between the primed C-terminal tail of ubiquitin and Ubc13”. Please clarify this.

The C-terminus of ubiquitin is not primed in our structures. We have corrected the legend for Supplementary Figure 2f.

Supplementary Figure 2c: The authors have to define what type of electron density map is shown and at what contour level.

We have specified the electron density map details in the legend for Supplementary Figure 2e.

Fig 5b: The authors should label the ubiquitin residues in the right and middle panels.

The ubiquitin residues have been labelled.

Some of the color choices in the figures are not optimal when printed. Examples are:

- Fig 1a: Black on green is difficult to read
- Fig 2d: The two shades of blue indicating no E3 and the E144A mutant very similar and should be changed

The colours have been changed to improve clarity.

The authors should carefully check the manuscript for typos and errors. Some examples are:

- p. 5 l. 122 'symmetrical' should be symmetrical.
- p. 59 l. 222: 'Fig. S5a,b' should be Fig. 5a,b.
- Fig 5c: In the TRIM5a panel K80 should be R80 and R72 should be K72.
- Table S1: The actual PDB codes should be given in the header column instead of the words "PDB code".

These errors have been corrected and the entire manuscript has been carefully reviewed.

REVIEWERS' COMMENTS:

Reviewer #1 (Remarks to the Author):

The authors have satisfactorily addressed my concerns. I find the manuscript to be acceptable for publication.

Reviewer #2 (Remarks to the Author):

The authors have addressed all of my concerns.

Reviewer #3 (Remarks to the Author):

In the revised manuscript "The activity of TRAF RING homo- and heterodimers is regulated by Zinc Finger 1", Adams et al. have included additional data and discussions to greatly improve the manuscript. They have now clarified the role of Arg-link and strengthened their results on the formation of TRAF heterodimers. The authors have adequately addressed all my previous concerns and I recommend the revised manuscript for publication in Nature Communications.

A few minor considerations/corrections remain that the authors should address.

Fig 4d: The red 'X' symbols depicting the different mutations in some cases are a bit confusing. Especially in row 7 (TRAF6 Arg-link/TRAF5), the position of the 'X' appears more like that the mutation is in TRAF5. The authors might want move the symbol slightly upwards towards the TRAF6 molecule.

Supplementary Table 1: The values for the resolution are still not consistent. For both structures the high-resolution values differ between the overall resolution and the resolution for the highest resolution shell.

Below we address the editorial matters and minor comments from the referees.

Reviewer #1 (Remarks to the Author):

The authors have satisfactorily addressed my concerns. I find the manuscript to be acceptable for publication.

Reviewer #2 (Remarks to the Author):

The authors have addressed all of my concerns.

Reviewer #3 (Remarks to the Author):

In the revised manuscript “The activity of TRAF RING homo- and heterodimers is regulated by Zinc Finger 1”, Adams et al. have included additional data and discussions to greatly improve the manuscript. They have now clarified the role of Arg-link and strengthened their results on the formation of TRAF heterodimers. The authors have adequately addressed all my previous concerns and I recommend the revised manuscript for publication in Nature Communications.

A few minor considerations/corrections remain that the authors should address.

Fig 4d: The red ‘X’ symbols depicting the different mutations in some cases are a bit confusing. Especially in row 7 (TRAF6 Arg-link/TRAF5), the position of the ‘X’ appears more like that the mutation is in TRAF5. The authors might want move the symbol slightly upwards towards the TRAF6 molecule.

Supplementary Table 1: The values for the resolution are still not consistent. For both structures the high-resolution values differ between the overall resolution and the resolution for the highest resolution shell.

We have adjusted the position of the X symbols to ensure their meaning is clear. We have also made the requested corrections to Table 1.